# Hyper Mucinous Proliferations in the Mucosa of Patients with Inflammatory Bowel Disease: Histological Lesions with a Real Potential for Neoplastic Evolution?

**DOI:** 10.3390/diagnostics14050499

**Published:** 2024-02-26

**Authors:** Enrico Costantino Falco, Davide Giuseppe Ribaldone, Gabriella Canavese

**Affiliations:** 1Department of Pathology, Città della Salute e della Scienza di Torino, 10126 Turin, Italy; enricoc.falco@gmail.com; 2Department of Gastroenterology, Città della Salute e della Scienza di Torino, 10126 Turin, Italy; davrib_1998@yahoo.com

**Keywords:** inflammatory bowel disease, biopsy, hyper mucinous proliferation, biology, dysplasia

## Abstract

Background and Aims: Mucin disfunction is a critical event in the pathogenesis of inflammatory bowel disease (IBD). Although hyper mucinous conditions have a still debated implication in the clinical evolution of this disorder, hyper mucinous villous proliferations were found to have a preneoplastic biologic potential. We studied morphologic and immunophenotypic characteristics of these lesions in ileocolonic resections for IBD to add evidence about the evolutive potential of these lesions in samples with well oriented wall structures. Methods: Morphologic characteristics of bowel samples from 20 patients resected for IBD and with raised lesions at gross examination were studied and sections from cases with hyper mucinous lesions were stained with the following antibodies: Ki 67, p21, and p27, which were employed to evaluate the characteristics of the proliferative and differentiative activity of the epithelial structures; mismatch repair proteins and p53 have been studied as proteins implicated in carcinogenesis in IBD-affected mucosa; mucins subtypes in hyper mucinous structures were evaluated with MUC-2 and MUC-6. The results in 11 cases of saplings were that they harbored hyper mucinous proliferations. The occurrence of hyper mucinous structures was not related to dysplastic lesions, pseudo pyloric metaplasia, subtype of disease, or activity. In only one of our cases, mild cytologic atypia in the proliferative compartment was detected. Proliferation markers (Ki 67, p53) were expressed in the proliferative compartments of mucosal crypts and antiproliferative proteins p21 and p27 were expressed in differentiated epithelium. MMR proteins expression was limited to the proliferative compartment of the hyper mucinous projections. Mucin subtypes distribution was regular in the epithelium of hyper mucinous proliferations. Conclusions: The present monocentric retrospective study was conducted on surgical samplings with well oriented crypts. Collected data show that hyper mucinous features are frequent occurrences in raised lesions in IBD patients. In hyper mucinous proliferations of the selected cases, the status of the proliferative cycle, the expression of the proteins most frequently involved in carcinogenetic pathways of mucosa affected by IBD, and the mucins subtypes expression have no evident anomalies. Findings are not consistent with the increased risk of neoplastic evolution observed in other studies; rather, they suggest a hyperplastic nature. However, the capacity of hyper mucinous raised lesions for neoplastic evolution should be ruled out with more extensive prospective studies to identify functional defects that could explain the hypothesized neoplastic potential.

## 1. Introduction

Inflammatory bowel disease (IBD) is a chronic disorder with a complex and poorly understood pathogenesis. Mucins dysfunction plays an important role in IBD development, given that alterations in the protective mucinous layer that covers the intestinal mucosa lead to tissue damage and immune system activation [1,2]. A recent meta-analysis demonstrated a global increase in mucin secretion (mainly, MUC 2 and 3) in patients with IBD [3]. Evidence of modifications concerning mucins glycosylation is another recent acquisition in IBD biology [4]. However, further evidence should be acquired to establish the effective structural, functional, and distributive modifications of mucins during the different evolutive phases of the disease.

Interestingly, the occurrence of epithelial cells with mucins overproduction in colonic mucosa has recently been related to an increased risk of cancer development in patients with IBD. Rubio et al. proved that villous dysplasia, defined as epithelial proliferations characterized by villous structure covered with a continuous row of goblet cells (enteric cells with a large mucinous vacuole) and some degree of cytologic atypia in the bottom of the crypts, were statistically related to the occurrence of invasive adenocarcinoma [5]. In recent studies conducted on IBD patients with colorectal cancer (CRC), hyper mucinous dysplasia, a lesion with morphologic features similar to villous dysplasia, was listed among the non-conventional (i.e., non-adenomatous) dysplastic lesions. Hyper mucinous dysplasia was defined as a tubulo-villous or villous epithelial proliferation consisting of tall mucinous cells with typically mildly elongated, hyperchromatic nuclei with prominent mucinous differentiation representing >50% of the lesion, with mild cytologic atypia seemingly decreasing towards the surface of the villi. According to another study of the same group, hyper mucinous dysplasia, that represents the 9% of non-conventional dysplasia, is the subtype most frequently associated with high-grade dysplasia and cancer (57%) [6,7,8].

Given that disfunction in mucin production is a well-recognized feature of tissue damage in the colonic mucosa of patients affected by IBD, we believe that a standardized classification of lesions with hyperproduction of mucus and the definition of their biological potential is fundamental for the pathologist’s contribution to the patient’s follow-up.

Since the available data on these lesions come from biopsy sampling [6,7,8], with the consequent limitations due to tissue dissociation, superficiality of sampling, and inappropriate section plane, we planned to analyze mucosal hyper mucinous proliferations, defined as raised lesions with continuous rows of enterocytes with high mucin content in the epithelial lining, in a cohort of ileocolonic surgical resections for IBD, where a good orientation of the tissue is more easily obtained. Moreover, we conducted an immunohistochemical characterization on these lesions to explore the functional characteristics of the epithelium that covers hyper mucinous proliferations.

At first, we assessed the immunohistochemical expression of cell cycle markers and cell cycle regulators. Ki 67, a marker of S-phase, was employed to define the extension of the proliferative district of the hyper mucinous villous projections [9] while the cyclin-dependent kinases (CDK) p21 (activated by native p53) and p27, two proteins involved in the arrest of the cell cycle and in the promotion of differentiative activity, were adopted to evaluate maturation processes in the hyper mucinous epithelium [10,11,12]. With the same purposes, we analyzed the expression of the main cell cycle regulators involved in carcinogenesis in bowel mucosa affected by IBD. Overexpression of p53, one of the main regulators of the cell cycle, is an early event in IBD-related carcinogenesis [13,14,15]. Mismatch repair (MMR) proteins restore the replication mistakes of microsatellite sequences (non-codifying, repeated regulatory sequences of DNA) and loss of function of these proteins results in a hyper mutational condition promoting cancer development and represents a genomic defect frequently involved in colorectal cancer development, even in IBD-related cancers [16,17,18,19].

On the other side, we planned to define the immunophenotyping of mucin stored in hyper mucinous cytoplasm, in particular, the expression of MUC 2, the most diffuse mucin in the colonic epithelium, and MUC 6, a mucin mostly secreted in the gastric epithelium [20,21,22].

## 2. Materials and Methods

A series of 187 consecutive surgical resections for IBD performed in our institution, a reference center in Piedmont, Italy, from January 2019 to July 2020, were considered in the study and the 20 cases in which raised pseudo polypoid lesion were observed at gross examination were selected for the survey. The most relevant clinical data were collected for all the patients.

All the histologic material (305 samplings; mean n° of samples/case: 15.3) was revised by two gastroenteropathologists (G.C., E.F.). In the histological specimens, the hyper mucinous raised lesions were identified as “epithelial projections with a continuous layer of tall mucinous cells, with no or scanty interspersed enterocytes”, as defined by the authors who described hyper mucinous dysplasia [6,7].

In the cases where these changes were detected, the number of involved samples was calculated and the cytological characteristics of the epithelial structures and the association with dysplastic and metaplastic changes were assessed.

One or two of the best oriented and most representative samples from every case with hyper mucinous raised lesions were selected and a series of immunohistochemical staining was performed on sections from the selected blocks. The following antibodies were employed: KI 67 Mib-1—anti-human Ki-67 monoclonal antibody (SP6); p21—anti-p21/WAF1/Cip1 (DCS-60.2) (DCS-60.2) monoclonal antibody; anti-p27 KIP 1 polyclonal antibody (SX53G8); p53—anti-p53 monoclonal antibody (DO7); PMS2—anti-PMS2 (MRQ-28); MLH-1—MLH1 monoclonal antibody (ES05); MSH2—anti-MSH-2 monoclonal antibody (25D12); MSH6—anti-MSH-6 monoclonal antibody; MUC2—anti-Mucin 2/MUC2 monoclonal antibody (F-2); MUC6—anti-Mucin 6/MUC6 monoclonal antibody (CLH5).

Portions of morphologically regular mucosa present in the selected samples were used as internal controls both for markers relating to the proliferative cycle and for markers of mucin subtypes.

## 3. Results

### 3.1. Cohort Characteristics

The clinical characteristics (subtype of IBD, type of resection, mean age, and disease duration) of the 20 selected cases of colonic and ileocolonic resection for IBD including raised lesions are summarized in Table 1.

The protruding lesions were coated by an epithelium with hyper mucinous changes (adopting the criteria defined in Section 2) in 11 of the 20 cases (55%) (Figure 1), while the mean rate of the involved samples for a single case was 36.5%. In 9 cases (45%), the hyper mucinous proliferations were associated with hyperplastic/regenerative changes with mild nuclear atypia and mucin depletion. Data about the distribution of IBD subtypes, occurrence of associated conventional (adenomatous) dysplasia, existence of gastric metaplasia (a benign reactive change associated with chronic disease), and grade of activity (in all cases, active disease was detected, except for 1 case in the hyper mucinous group) were matched in the two groups (see Table 2). Associated dysplasia had high-grade features in the case of the group with hyper mucinous changes (Figure 2) and low-grade in the case of the other group (residual low-grade dysplasia after a previous resection for cancerous polyp—pT1 adenocarcinoma).

### 3.2. Morphologic and Immunophenotypic Characteristics of Hyper Mucinous Lesions

Following the above-described criteria, the hyper mucinous raised lesions detected in our investigation were structured as sessile or semi-peduncolated tubular structures with a linear or tortuous, irregular axis, and only in a portion of the detected epithelial bulge had an obvious villous profile. Epithelial projections had a regular basal crypt, except there was focal architectural irregularity in 4 cases (36.4%) and the crypt epithelium contained Paneth cells in ileal samplings.

The cytologic features of the basal portion of hyper mucinous projections were defined as follows: in most of the cases (8/11 cases; 72.7%), slight nuclear crowding with minimal chromatin condensation was observed (Figure 3A) (1 of these was associated with an adenomatous polyp with high-grade dysplasia); nuclear packing was associated with overt hyperchromasia in 1 case (9.1%); while obvious nuclear irregularity consistent with mild atypia (chromatin condensation, membrane thickening or slight increase in mitotic figures) was observed in another case (9.1%) (Figure 3B).

In 8/11 (72.7%) cases, Ki67-Mib-1 immunostained the nuclei in the basal zone of the villous projections, roughly corresponding to the crypt of the ordinary small intestinal villi. In 3/11 (27.3%) cases, a slight expansion of the stain towards the mucosal surface, always within the lower third of the villous projections, was observable, including the case with mild nuclear atypia. Also, p21 Waf-1 stained the nuclei of the epithelium lining the villous projections of all the cases, with a reciprocal distribution with respect to Ki67 (Figure 4 and Figure 5). A faint staining, with a similar distribution, was observed in all cases (11/11) with p27 Kip1 staining (Figure 6D). Also, p53 staining was observed in 4 cases, and the protein was co-expressed along with ki67 in the deep portion of the villous structures (Figure 5C).

In all cases (11/11), the proteins of the mismatch repair (MLH-1, MSH2, MSH6, and PMS2) were expressed in the basal compartment of the hyper mucinous projections, reproducing Ki67 Mib-1 staining (Figure 6).

The expression of MUC-2 had the same distribution in all the studied cases (11/11): a strong staining in the extra vacuolar cytoplasm of mucinous cells was observed (Figure 5D and Figure 7A). The hyper mucinous epithelium was negative for MUC6, which stained, instead, the cytoplasm of the metaplastic gastric epithelium (pseudopyloric metaplasia) when present in the mucosa with hyper mucinous projections (3/11 cases) (Figure 7B).

In the case harbouring high-grade dysplasia, data are related to the residual hyper mucinous, non-dysplastic component.

## 4. Discussion

The detection of preneoplastic lesions is one of the most important tasks of the pathologists in follow-up of patients with long-course inflammatory bowel disease, and we think that maximum effort should be made to define the biology and morphological characteristics of these lesions. Even if the available data about the effective role of mucins in the paradoxical inflammatory response that characterizes the disease are, on the whole, contradictory, the occurrence of mucosal villous proliferation with a hyper mucinous epithelium (hyper mucinous villous dysplasia) was found to be associated with the development of neoplastic lesions and, therefore, included among the non-conventional preneoplastic lesions [6,7,8].

With the aim of improving evidence about the neoplastic potential of such lesions, we analyzed some of the most significant morphological and biological features of raised mucosal lesions with a hyper mucinous epithelium in histological samplings from a consecutive series of surgical resections for IBD of a single center. According to our knowledge, our retrospective analysis is the first investigation on the topic conducted on samples from surgical specimens where adequate orientation and consequent evaluation of the proliferative and differentiative compartments of crypts is easier than in biopsy samplings.

The first key result of the study was the high frequency of raised lesions with a hyper mucinous epithelium; lesions with these features were detected in more than half of the cases of our cohort. A recent survey on non-conventional mucosal lesions in IBD patients reported a high frequency of these lesions, even if the study considered serrated features that were not included in the selection criteria of our study [19].

No significant correlation at chi-square test was observed between hyper mucinous proliferations and the occurrence of conventional dysplastic lesions, the subtype of disease, the grade of activity, and the occurrence of gastric metaplasia.

In our survey, the hyper mucinous lesions were characterized by villous or tubulo-villous projections covered by a layer of tall enterocytes with inconspicuous small nuclei and prominent mucinous cytoplasmic vacuoles with a “foveolar” texture. The cytologic characteristics of the epithelial elements showed negligible nuclear irregularity, and only in one case, the nuclear characteristics of the elements of the deep portion of hyper mucinous structures showed nuclear changes compatible with mild atypia.

The proliferative compartment of hyper mucinous structures, as defined by Ki67 immunostaining, was confined to the lower third of the villous projections, reproducing the ordinary distribution of the elements in the replicative phase of the colonic crypts. On the other side, a decreased expression of p21 Waf1, a cyclin-dependent kinase with an inhibitory function on the proliferative cycle, was related to the risk of neoplastic development in IBD [12]. In all cases of our cohort, the protein was regularly distributed to the nuclei on the superficial 2/3 of the villous projections of the epithelium, i.e., in the differentiative compartment of these structures; given the faint expression of p27Kip-1, we could argue the prominent role of p21 Waf1 in cell cycle suppression in the differentiated hyper mucinous enterocytes of the villi.

Interesting suggestions were provided by the immunostaining of some of the proteins most frequently involved in carcinogenetic pathways of mucosa affected by IBD. We noted that p53, the main regulator of the proliferative cycle in the epithelial layer of the enteric mucosa, demonstrated a significant role in IBD-related carcinogenesis; the protein was frequently mutated in IBD-affected mucosa, and earlier than the development of evident dysplastic lesions [15], following a model of inflammation-dependent carcinogenesis (field cancerization) [23]. In our cases, p53 expression was observable in 4 cases and the protein was co-expressed with ki67 in basal proliferative portions of hyper mucinous villi, showing a pattern of distribution more attributable to hyperplastic lesions than to dysplastic ones. [24]. Mismatch repair (MMR) proteins expression was maintained in the proliferating cells of hyper mucinous villous structures; the finding implies the absence of microsatellite instability, a loss of function frequently involved in colorectal carcinogenesis and recently described in IBD-related neoplasms [17,19].

In all our cases, cells lining the hyper mucinous epithelium synthesized mucin gene product MUC2, like the goblet cells of intestinal mucosa [20,21]. The finding suggests that hyper mucinous proliferations could represent an aspect of the remission phase of the disease, given the hypothesis of an inverse relation between the activity score and mucin production [3]. MUC6 was regularly expressed in the gastric epithelium in the areas of pseudo pyloric change, a metaplastic modification of IBD bowel mucosa frequently associated with chronic disease.

## 5. Conclusions

In conclusion, data collected from this monocentric retrospective survey, although carried out on a limited number of cases, prove that (1) raised hyper mucinous lesions are frequent events in patients with IBD and (2) the morphologic features of the proliferative basal portion of the hyper mucinous proliferations, the status of the proliferative cycle regulators, and the expression of the proteins more strictly related with cancer risk of these epithelial proliferations have no evident anomalies and are not consistent with an in-creased risk of neoplastic evolution.

Collected data suggest that hyper mucinous proliferations could represent hyperplastic reactive proliferations, that are a common feature of bowel mucosa in the chronic phase of IBD in regenerative mucosa. The hyper mucinous proliferations included in this study lack nuclear changes, even with severe atypia, of the basal proliferative portion of the crypts, described as characteristic of hyper mucinous dysplasia in a previous large study on nonconventional dysplastic lesions [6,7] and frequently observed in regenerative changes of IBD-affected mucosa, causing the well-known issues of differential diagnosis versus conventional dysplasia [25,26,27].

The contribution of our study to the evolution of the diagnostics of preneoplastic le-sions of patients with IBD can be resumed in two acquisitions: the lesions with hyper mu-cinosis have proven to be a frequent occurrence in bowel mucosa and further studies are necessary to identify more precisely the morphological, immunophenotypic, and molecu-lar characteristics correlated to any potential for neoplastic evolution; on the other side, the sizes of the histological samplings and their handling in the laboratory could represent an important variable in the evaluation of these lesions.

## Figures and Tables

**Figure 1 diagnostics-14-00499-f001:**
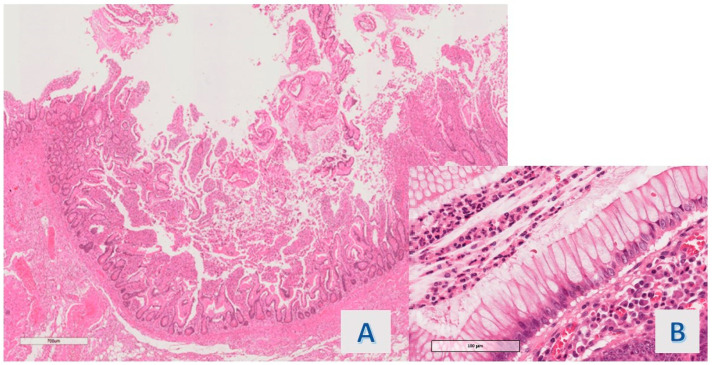
Case 2: Characteristics of hyper mucinous structures. (**A**) villous projections (O.M. 200×), (**B**) cytology of hyper mucinous epithelium, with continuous rows of goblet cells (O.M. 40×).

**Figure 2 diagnostics-14-00499-f002:**
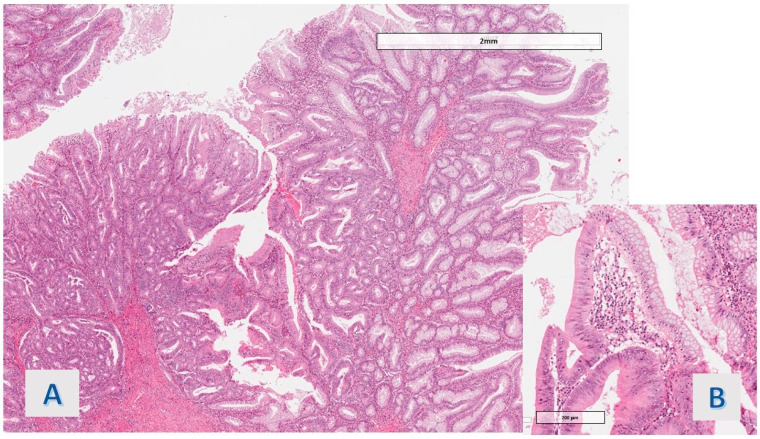
Case 1: Transition area between dysplastic adenomatous villous structures and hyper mucinous villous structures. (**A**) hem eos (O.M. 40×), (**B**) hem eos (O.M. 200×).

**Figure 3 diagnostics-14-00499-f003:**
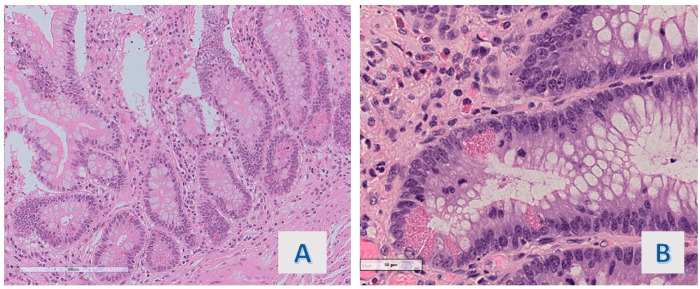
(**A**) Case 2: Basal portion of hyper mucinous projection (O.M. 100×). (**B**) Case 18: Basal portion of hyper mucinous projection with mild cytocariologic irregularities (O.M. 400×).

**Figure 4 diagnostics-14-00499-f004:**
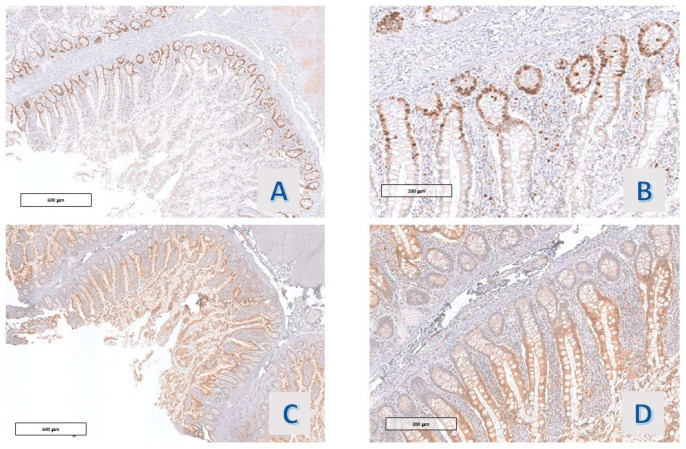
Case 9: (**A**,**B**) Ki67 Mib-1 (O.M. 40× and 200×), (**C**,**D**) p21 (O.M. 40× and 200×).

**Figure 5 diagnostics-14-00499-f005:**
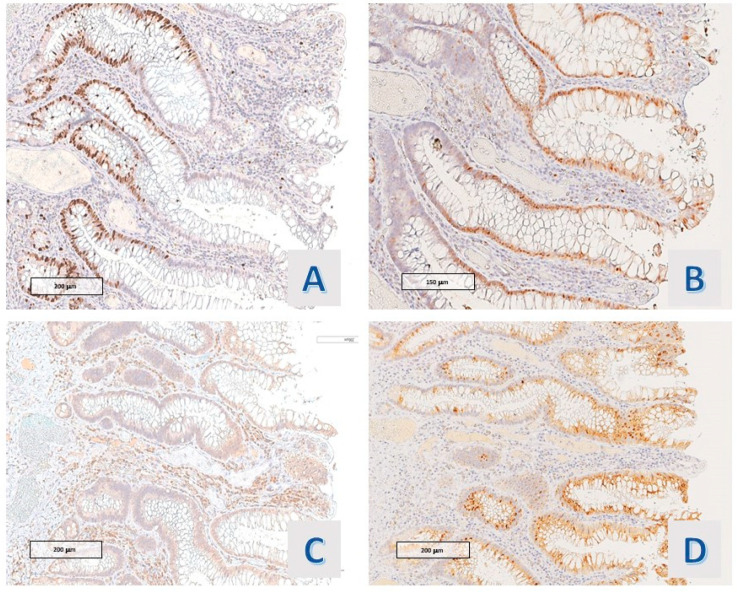
Case 18: Hyper mucinous projections with mild cytologic atypia. (**A**) KI67, (**B**) p21, (**C**) p27, and (**D**) MUC 2 (O.M. 100×).

**Figure 6 diagnostics-14-00499-f006:**
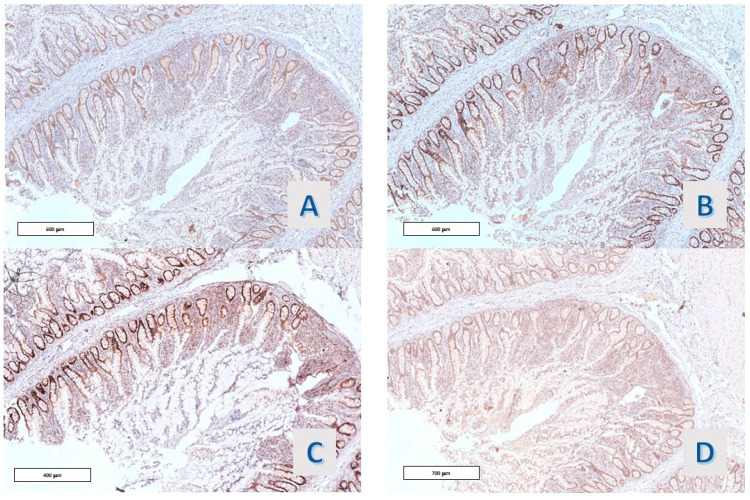
Case 9: (**A**) PMS2, (**B**) MSH2, (**C**) MSH6, and (**D**) MLH1 (O.M. 100×).

**Figure 7 diagnostics-14-00499-f007:**
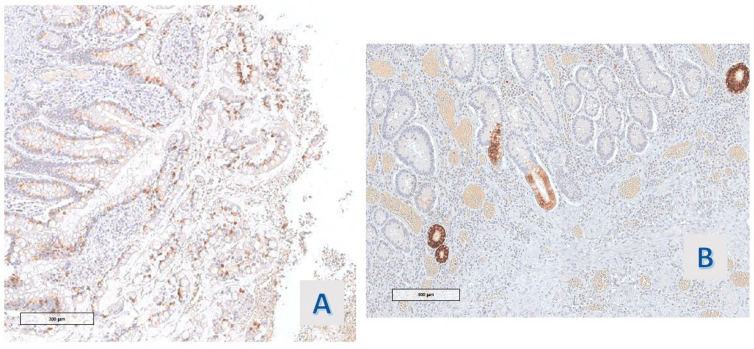
Case 9: (**A**) Muc-2 and (**B**) Muc-6 (O.M. 200×).

**Table 1 diagnostics-14-00499-t001:** Clinical characteristics of selected cases.

Parameters	Count of the Cohort (*n*: 20)
subtype of IBD (%)	
Crohn’s disease	11 (55)
ulcerative colitis	9 (45)
type of resection (%)	
colectomy	7 (35)
right hemicolectomy.	7 (35)
segmental resection	3 (15)
extension of previous resection.	3 (15)
mean age (range)	53.4 (24–83)
reason for resection	
non-responsive	8 (40)
high-grade dysplasia at biopsy	2 (10)
occlusion	2 (10)
fistula/abscess	2 (10)
other/NA	6 (30)
Mean duration of the disease (range)	12.15 (1–29)

NA: not available.

**Table 2 diagnostics-14-00499-t002:** Main histological parameters occurrence in selected cases.

Parameters	With Hyper Mucinous Raised Lesions *n* (%) (Total = 11)	Without Hyper Mucinous Raised Lesions *n* (%) (Total = 9)	X^2^ Statistics
subtype of IBD (*n*)			
Crohn’s disease (11)	6 (55)	5 (45)	0.963969 not significant at *p* < 0.05
ulcerative colitis (9)	5 (56)	4 (44)	
pseudopiloric (gastric) metaplasia (*n*)			
present (4)	3 (75)	1 (35)	0.736042 not significant at *p* < 0.05
absent (16)	8 (50)	8 (50)	
associated dysplasia (*n*)			
present (2)	1 (50)	1 (50)	0.880905 not significant at *p* < 0.05
absent (18)	10 (56)	8 (44)	
activity (*n*)			
present (1)	1 (50)	0 (0)	not assessable (cell value = 0)
absent (19)	10 (56)	9 (100)	

## Data Availability

The data underlying this article are available in a repository of our institution and will be shared on reasonable request to the corresponding author.

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
