# Peer review of "Hyper Mucinous Proliferations in the Mucosa of Patients with Inflammatory Bowel Disease: Histological Lesions with a Real Potential for Neoplastic Evolution?"

_diagnostics, 2024, doi:10.3390/diagnostics14050499_

Round 1

Reviewer 1 Report

Comments and Suggestions for Authors

Article is difficult to understand its content.

Discussion is poor, and Introduction does not explain fully why authors examine this problem.

According to refernces, only 7 refrences have published in 2019 year or later. The remaining 22 cited articles are older, and some of them too old.

The amount of cases is veroo small.

Comments on the Quality of English Language

Manuscript requires minor English editing

Author Response

Author's Reply to the Review Report (Reviewer 1)

Many thanks for your interest in our work and for your willingness to review the manuscript,

  1. Article is difficult to understand its content.

In fact, the topic is controversial and still little explored, but we believe that it is important to contribute to clarifying the prognostic significance of these lesions, given their frequency in the mucosa of the intestine affected by IBD, as our study has shown. We hope that the changes made in the introduction section (lines 66-69) and in the conclusions section (lines 309-315) can make the utility of the study more clear.

  1. Discussion is poor, and Introduction does not explain fully why authors examine this problem.

See point 1

  1. According to refernces, only 7 refrences have published in 2019 year or later. The remaining 22 cited articles are older, and some of them too old.

It’s true. We think that this reveals the limitations and poor updating of scientific research on the topic. Naturally, some studies may have escaped our bibliographic research on the topics covered. We are available to review the literature, if so.

  1. The amount of cases is veroo small.

In designing this study, we chose to ensure a comprehensive immunophenotyping study, limiting the number of cases selected.

Reviewer 2 Report

Comments and Suggestions for Authors

In this study, author investigated the hyper mucinous proliferations in mucosa of patients with IBD. They studied morphologic and immunophenotypic characteristics of these lesions in ileocolonic resections for IBD to add evidence about evolutive potential of these lesions in samples with well oriented wall structures. They assessed the immunohistochemical expression of cell cycle markers and cell cycle regulators. They found that, in hyper mucinous proliferations of the selected cases the status of proliferative cycle, the expression of the proteins most frequently involved in carcinogenetic pathways of mucosa affected by IBD and the mucins subtypes expression have no evident anomalies. In general, this paper is interesting and well-conducted. Here are some comments from this reviewer:

1. For the immunostaining data, there are no control samples which were used for comparison.

2. Scale bar should be added for all the images.

3. For figure 2A, scale bar should be inverted.

4. How about the lgr5+ stem cells expression? If the colon tissues were hyper-proliferated, lgr5+ stem cells should be increased.

Author Response

Many thanks for your interest in our work and for your willingness to review the manuscript.

In this study, author investigated the hyper mucinous proliferations in mucosa of patients with IBD. They studied morphologic and immunophenotypic characteristics of these lesions in ileocolonic resections for IBD to add evidence about evolutive potential of these lesions in samples with well oriented wall structures. They assessed the immunohistochemical expression of cell cycle markers and cell cycle regulators. They found that, in hyper mucinous proliferations of the selected cases the status of proliferative cycle, the expression of the proteins most frequently involved in carcinogenetic pathways of mucosa affected by IBD and the mucins subtypes expression have no evident anomalies. In general, this paper is interesting and well-conducted. Here are some comments from this reviewer:

  1. For the immunostaining data, there are no control samples which were used for comparison.

In fact, the controls used for the study were not described in the materials and methods. we used the crypts of regular portions of mucosa present in the sample as internal controls. We added the information in Material and methods (line 120-122)

  1. Scale bar should be added for all the images.
  2. For figure 2A, scale bar should be inverted.

Sorry for the inaccuracy. Scale bars were revised and added when absent.

  1. How about the lgr5+ stem cells expression? If the colon tissues were hyper-proliferated, lgr5+ stem cells should be increased.

The proposal is undoubtedly interesting. To our knowledge, LGR5 enhances Wnt/β-catenin signalling. We chose not to include the evaluation of beta-catenins in the study because, even if their role in the carcinogenesis of sporadic colon carcinoma is well established, its implication in IBD-related carcinoma appears to us to be more dubious and less documented. Consequently, we did not consider beta-catenin-regulators. However, our knowledge about LGR5 is limited, and we are available to discuss the proposal.

Round 2

Reviewer 1 Report

Comments and Suggestions for Authors

Authors reffered to suggestions. However, changes would require total modifications at the stage of planning.

Reviewer 2 Report

Comments and Suggestions for Authors

Authors addressed all my questions. No more comments